



# NITROUS OXIDE (N₂O) in MACQUARIE HARBOUR, TASMANIA

Maxey, Johnathan Daniel[1,2], Neil D. Hartstein[2], Hermann W. Bange[3], Moritz Müller[1]

[1]Faculty of Engineering, Computing and Science, Swinburne University of Technology, Kuching 93350, Malaysia
[2]ADS Environmental Services, Kota Kinabalu, Sabah, 88400, Malaysia
[3]GEOMAR Helmholtz Centre for Ocean Research Kiel, Wischhofstr. 1-3, 24148 Kiel, Germany

*Correspondence to*: Johnathan Daniel Maxey, Neil D. Hartstein, Hermann W. Bange, and Moritz Müller

**Abstract.** Fjord-like estuaries are hotspots of biogeochemical cycling due to steep physicochemical gradients. The spatiotemporal distribution of nitrous oxide ($N_2O$) within many of these systems is poorly described, especially in the southern hemisphere. The goal of this study is to describe the spatiotemporal distribution of $N_2O$ within a southern hemisphere fjord-like estuary, describe the main environmental drivers of this distribution, the air/sea flux of $N_2O$, and the main drivers of $N_2O$ production. Cruises were undertaken in Macquarie Harbour, Tasmania to capture $N_2O$ concentrations and water column physicochemical profiles in winter (July 2022), spring (October 2022), summer (February 2023), and autumn (April 2023). $N_2O$ samples were collected at one depth at system end members, and at 5 depths at 4 stations within the harbour.

Results indicate that $N_2O$ is consistently supersaturated (reaching 170% saturation) below the system's freshwater lens where oxygen concentrations are often hypoxic, but infrequently anoxic. In the surface lens, levels of $N_2O$ saturation vary with estimated river flow and with proximity to the system's main freshwater endmember. The linear relationship between AOU and $\Delta N_2O$ saturation indicates that nitrification is the process generating $N_2O$ in the system. When river flow was high (July and October 2022), surface water $N_2O$ was undersaturated (as low as 70%) throughout most of the harbour.

When river flow was low (February and April 2023) $N_2O$ was observed to be supersaturated at most stations. Calculated air/sea fluxes of $N_2O$ indicated that the system is generally a source of $N_2O$ to the atmosphere under weak river flow conditions and a sink during strong river flow conditions. The diapycnal flux was a minor contributor to surface water $N_2O$ concentrations, and subhalocline $N_2O$ is intercepted by the riverine surface lens and transported out of the system to the ocean during strong river flow conditions. In a changing climate, Western Tasmania is expected to receive higher winter rainfall and lower summer rainfall which may augment the source and sink dynamics of this system by enhancing the summer / autumn efflux of $N_2O$ to the atmosphere.

This study is the first to report observations of $N_2O$ distribution, generation processes, and estimated diapycnal / surface $N_2O$ fluxes from this system.



## 1. Introduction

Despite the fact that fjords and fjord-like estuaries represent only a small portion of the coastal area worldwide they are responsible for sequestering 11% of the global organic carbon (C) burial along terrestrial margins (**Smith *et al.,* 2015**; **Bianchi *et al.,* 2018, 2020**). These systems are significant sources of greenhouse gasses (GHG) to the atmosphere (**Wilson *et al.,* 2020**; **Rosentreter *et al.,* 2023; Bange *et al.,* 2024**). Many are heavily stratified with strong water column physicochemical gradients (**Acuña-González *et al.,* 2006; Inall and Gillibrand, 2010; Hartstein *et al.* 2019; Salamena *et al.,* 2021, 2022; Maxey *et al.* 2022**). These gradients can be influenced by mesoscale climate drivers like NAO and SAM (see **Austin and Inall 2002; Gillibrand *et al.,* 2005; Maxey *et al.,* 2022**) and local scale drivers like fresh water input and marine intrusions (**Inall and Gillibrand 2010; Hartstein *et al.,* 2019; Maxey *et al.,* 2020; Salamena *et al.,* 2022**).

Nitrous oxide ($N_2O$) is a potent GHG whose increased presence in the atmosphere is primarily driven by emissions from agricultural soils with an increased presence in poorly oxygenated marine systems (**Laffoley and Baxter 2019**; **Ji *et al.,* 2020; Wilson *et al.,* 2020; Wan et al., 2022; Orif *et al.,* 2023**). With a global warming potential nearly 300 times that of $CO_2$ (**Myhre *et al.,* 2013**; **Etminan *et al.,* 2016; Eyring *et al.,* 2021; Forster *et al.,* 2021**) $N_2O$ is a key focus of climate studies especially regarding ozone layer depletion. $N_2O$ is a precursor to NO, and a major ozone depleting substance in the atmosphere (**Nevison and Holland, 1997**; **Ravishankara *et al.,* 2009; Portmann *et al.,* 2012**). $N_2O$ production occurs through the microbially mediated processes of ammonia oxidation, nitrite ($NO_2^-$) reduction, and nitrate ($NO_3^-$) reduction (**Kuypers *et al.,* 2018**). $N_2O$ production in marine systems is governed by environmental conditions such as dissolved oxygen (DO) availability, ammonium ($NH_4^+$) availability, light availability, temperature (*e.g.* **Raes *et al.,* 2016**), pH (*e.g.* **Breider *et al.,* 2019**), and microbial community composition (*e.g.* **Wu *et al.* 2020**).

Estuarine systems have disproportionately high biological productivity relative to other marine systems (**Walinsky *et al.,* 2009; Gilbert *et al.,* 2010; Bianchi *et al.,* 2018, 2020**). This also applies to $N_2O$ dynamics with approx. 33% of marine $N_2O$ emissions coming from estuaries (**Bange *et al.,* 1996; Seitzinger *et al.,* 2000**; **Murry *et al.,* 2015**; **Reading, 2022; Rosentreter *et al.,* 2023**). Estuaries can act as net sinks (**Maher *et al.,* 2016; Wells *et al.,* 2018**) and sources (**De Bie *et al.,* 2002; Zhang *et al.,* 2010; Sánchez-Rodríguez *et al.,* 2022**) of $N_2O$ depending on factors controlling air/sea fluxes, waterbody/atmospheric concentrations (**Wells *et al.,* 2018**; **Bange *et al.* 2019**), land use modification (**Reading *et al.,* 2020; Chen *et al.,* 2022**), and even the presence of microplastics (**Chen *et al,* 2022**). Despite the advancements made thus far, our understanding of marine $N_2O$ distribution and atmospheric emissions needs improvement (**Bange *et al.,* 2024**), especially in southern hemisphere fjord-like systems (**Yevenes *et al.,* 2017**). Much of the current uncertainty lies with a lack of in-situ data describing seasonal $N_2O$ dynamics to constrain global emissions models (**Bange *et al.,* 2019**).

The purpose of this study was (1) to investigate the distribution and seasonal variability of $N_2O$ concentrations and emissions in a southern hemisphere fjord-like estuary and (2) to decipher the major physical and biological drivers of the $N_2O$ emissions.



## 2. Methods

### 2.1 Study Area

Macquarie Harbour is a southern hemisphere fjord-like estuary located on Tasmania's west coast (**Figure 1**). The harbour is oriented NW by SE, and is approximately 33 km long, 9 km wide, with a surface area of 276 km$^2$. The mouth of the harbour is constricted by a shallow (4-8m), long (14km) sill known as "Hells Gates". Hells Gates muffles tidal forcing resulting in harbour water levels primarily determined by river flow and wind set up (**Hartstein et al., 2019**). The morphology of this system results in sharp gradients of DO, salinity, and temperature which are seasonally dependant (**Creswell et al., 1989**; **Hartstein et al., 2019**; **Maxey et al., 2022**). In surface waters DO concentrations are nearly always in equilibrium with the air but decrease sharply through the halocline (~8m to 15m). Subhalocline layers (~15m to a few meters from the bottom) are observed to be below 62.5 µM more than 50% of the time (*see* **Maxey et al., 2022**). Near the seabed, episodic marine intrusions (deep water renewal) refresh the supply of DO. In the upper reaches of the harbour marine intrusions are much less common (*see* **Hartstein et al., 2019**; **Maxey et al., 2022**). In these areas the DO concentration falls below 31 µM nearly a third of the time (**Maxey et al., 2022**).

Like many fjord-like estuaries, the distribution of DO is driven by multiple physical and biological processes whose relative importance depends on position along the estuarine axis as well as water column depth (**Hartstein et al., 2019** and **Maxey et al., 2017, 2020, 2022**). There is almost no DO produced below the halocline (8m to 12m deep) as the overlying freshwater lens is high in chromophoric dissolved organic matter (CDOM) limiting light available to primary producers in the surface water layers (within 3m) (**Maxey et al., 2017, 2020**). Basin water oxygen concentrations and salinity are largely influenced by advection of marine water over Hells Gates (**Andrewartha and Wild-Allen 2017; Hartstein et al., 2019; Maxey et al., 2022**). This process is driven by low atmospheric pressure, sustained NW winds, and low catchment rainfall which itself is influenced by Southern Annular Mode (SAM) (**Hartstein et al., 2019; Maxey et al., 2022**). Hydrodynamic and oxygen tracer numerical simulations of the harbour by **Andrewartha and Wild-Allen (2017)** support measurements by **Maxey et al. (2022)** which show that basin water residence times and basin water oxygen concentrations are tied to the flow conditions of the freshwater endmembers. In their model basin water tracer concentrations were reduced by half in approximately 65 days during low river flow conditions (approximately 70 days at the surface) and in approximately 110 days during normal flow conditions (approximately 40 days at the surface).

The main source of freshwater to the harbour is located on its southeast end (the Gordon River) and drains a nearly pristine catchment (including the Franklin River) of approximately 5,682 km$^2$ (**Macquarie Harbour Dissolved Oxygen Working Group, 2014**; **Fig 1**). The Gordon River discharges an estimated 180,000 tons organic carbon (OC) per year (**Maxey et al., 2020, 2022**) into the estuary. It should be noted that this area receives the some of the highest rainfall (more than 2,500 mm year$^{-1}$) volume in Australia (**Dey et al., 2019**). The King River, located on the harbour's northern end, is the second largest contributor of fresh water to the estuary and drains a catchment area of 802 km$^2$. Unlike the Gordon River, the King River has a history of receiving treated mining (*e.g.* copper) effluent and transporting this to the harbour (**Carpenter et al., 1991; Teasdale et al., 2003**).



**Figure 1: Macquarie Harbour, Tasmania. Water sampling stations shown with red circles; Cape Grim Air Pollution monitoring station shown as a green star (see inset map). Cape Sorell Weather Station shown as an orange star. Gordon Above Denison stream gauge shown as a red star (see inset map). Aquaculture lease boundaries are shown as hollow rectangles. Lease locations are sourced from Land Information Systems Tasmania (LISTmap - https://maps.thelist.tas.gov.au/). Station names reflect general harbour locations where KR1 indicates King River 1; C10 and C08 indicate Central Harbour 10 and 08 respectively; WH2 indicates World Heritage Area 2; and GR1 indicates Gordon River station 1. Coordinates are displayed in GDA_1994_MGA_Zone_55. Bathymetry through the system shown as a dashed line, note that this track excludes stations KR4 and KR1.**



### 2.2    Experimental Design

$N_2O$ distribution was assessed by collecting water samples across 7 stations, including the harbour's endmembers (mouths of the Gordon and King Rivers as well as the harbour mouth at Hells Gates Inlet; *see* **Figure 1** and **Table 1**) and stations along the longitudinal axis of the harbour where the deepest basins are located (named KR1, C10, C08, and WH2). Samples collected at endmember stations were collected from a single depth as these stations are shallow. Samples in the harbour body were collected at 5 depths from the

surface (2m) to approx. 1m from the seabed. Collection campaigns were conducted in July 2022, October 2022, February 2023, and April 2023. At each station and depth three replicate vials (n = 3) were collected for the determination of $N_2O$ concentration.

### 2.3    Field Sampling

At each station, water quality sonde profiles were collected from the surface to the seabed at 1 meter intervals using a YSI EXO-1 equipped with optical DO (accuracy from 0 to 625 µM ± 3 µM or 1% of reading whichever is greater; precision is 0.03 µM), salinity (accuracy ± 0.1 or 1% of reading whichever is greater; precision is 0.01), temperature (accuracy is ± 0.15 ℃; precision is 0.01 ℃), and depth sensors. Sonde calibration was checked and corrected (when needed) each sampling period.

Water samples were collected at various depths (see **Table 1**) using a 5 L Niskin bottle sampler. Water sample parameters include dissolved Total Ammoniacal N ($NH_3 + NH_4^+$) (TAN), $NO_3^-$, and $N_2O$. $N_2O$ samples were collected in triplicate immediately after retrieval of the Niskin bottle by transferring water from the bottle through silicone tubing into a 20 mL borosilicate vial. Sample water was added to the vial by placing the tubing

at the bottom and allowing the vial to overflow several volumes before sealing with a butyl rubber stopper and aluminium crimp. After ensuring the sample vial is bubble free, 50 µL of saturated mercury chloride ($HgCl_2$) solution was injected into the sample to arrest biological activity. All $N_2O$ samples were shipped to GEOMAR in Kiel, Germany for analysis. Samples were measured in July/August 2023 within 4 to 12 months after sampling and were not affected by the storage time (**Wilson *et al.,* 2018**).

Water collected for dissolved inorganic nitrogen (N) was filtered immediately using 0.45 µm polyethersulfone syringe filters (Whatman Puradisc). Samples were stored in a chilled dark container until being transported to Analytical Services Tasmania in Hobart, Australia for analysis. Dissolved Total TAN and $NO_3^-$ were analysed using a Lachat Flow Injection Analyser. TAN and $NO_3^-$ analyses used methods based on APHA Standard

methods (2005) 4500-$NH_3$ H (reporting limit 0.005 mg $L^{-1}$) and 4500 - $NO_3^-$ $L^{-1}$ (reporting limit 0.002 mg $L^{-1}$).

**Table 1: Sampling stations showing coordinates, parameters, and sampling depth (in meters).**

| Station | Station Depth (m) (MSL) | Dissolved Oxygen Salinity Temperature | $N_2O$ | TAN ($NH_3 + NH_4^+$) | $NO_3^-$ |
|---------|------------------------|---------------------------------------|--------|----------------------|----------|
| **HG3** 352484, 5325594 | 8 | *Every Meter* | *5m* | *5m* | *5m* |





| | | | | | |
|---|---|---|---|---|---|
| **KR4**<br>*365018, 5327550* | 3 | *1m* | *1m* | *1m* | *1m* |
| **KR1**<br>*361316, 5325972* | 36 | *Every Meter* | *2, 12, 20, 30, 35m* | *2, 12, 20, 30, 35m* | *2, 12, 20, 30, 35m* |
| **C10**<br>*363708, 5320464* | 44 | *Every Meter* | *2, 12, 20, 30, 42m* | *2, 12, 20, 30, 42m* | *2, 12, 20, 30, 42m* |
| **C08**<br>*365489, 5317238* | 47 | *Every Meter* | *2, 15, 25, 35, 45m* | *2, 15, 25, 35, 45m* | *2, 15, 25, 35, 45m* |
| **WH2**<br>*370218, 5309894* | 32 | *Every Meter* | *2, 12, 20, 25, 30m* | *2, 12, 20, 25, 30m* | *2, 12, 20, 25, 30m* |
| **GR1**<br>*377784, 5300603* | 12 | *Every Meter* | *10m* | *10m* | *10m* |

**2.4      Analysis of Rainfall and River Loading Estimation**

River loading and rainfall were analysed using methods presented in **Maxey *et al.* (2022)** where rainfall and stream gauge data were collected from the Gordon River catchment, Strathgordon rainfall gauge station and the Gordon Above Denison (GAD) stream gauge (**Figure 1**). The rainfall and flow metrics computed include the average daily rainfall over a 20-day period prior to sampling; total accumulated rainfall 20,10, 5, and 3 days

prior to sampling; estimated Gordon River flow into the estuary; and measured flow at the GAD stream gauge.

Gordon River flow was estimated by scaling daily rainfall to the size of the catchment and assuming a rainfall and runoff coefficient of 0.70 adopted from a neighbouring catchment with similar land cover, geology, and slope (**Willis, 2008**). Additional streamflow from Gordon River dam releases was estimated by subtracting

scaled rainfall contributions to river flow measured at the GAD stream gauge. This flow was added to the estimated runoff entering the harbour. Rainfall and flow data were provided by the Australian Bureau of Meteorology (BOM).

$NO_3^-$ and TAN loading was estimated my multiplying the measured concentration of each parameter at station

GR1 (*see* **Figure 1** and **Table 1**) by the estimated Gordon River flow.

**2.5      Analysis of Water Column $N_2O$ Concentrations, Air/Sea Flux, and Diapycnal Flux**

**2.5.1      Determination of $N_2O$ Concentrations**

Water samples were analysed for $N_2O$ using the static-headspace equilibration method followed by gas

chromatographic separation (HP Agilent 5890) and detection with an electron capture detector (ECD) as described in **Bange *et al*, (2019)**, **Bastian (2017)**, and **Kallert (2017)**. The concentrations of $N_2O$ in the samples was calculated with the following equation (**Equation 1;** *see* **Bange et al., 2006**):





**Equation 1**

$$C_{obs} = \frac{x^{'}PV_{hs}}{RTPV_{wp}} + X'\beta P$$

$C_{obs}$ is the concentration (nmol L$^{-1}$) of N$_2$O in the water sample; **x'** is the measured dry mole fraction of N$_2$O in the sample vial's headspace; $P$ is the ambient pressure set to 1 atm; $V_{hs}$ and $V_{wp}$ are the volumes of the headspace in the vial and water in the vial; $R$ is the gas constant; $T$ is the temperature during equilibrium; and $\beta$ is the solubility of N$_2$O (**Weiss and Price, 1980**). The mean relative error of the concentration values obtained was 2.4% (± 0.16).

### 2.5.2    Estimation of N$_2$O Air/Sea Fluxes and N$_2$O Saturations

N$_2$O air/sea fluxes ($F$ in µmol m$^{-2}$ d$^{-1}$) were estimated using equations from **Zhang et al., (2010)** and **Bange et al., (2019)** (**Equation 2**) *Where*:

**Equation 2**

$$F = K * (C_{obs} - C_{eq})$$

$C_{obs}$ is the measured concentration (nmol L$^{-1}$) of N$_2$O in the water sample; $C_{eq}$ is the air-equilibrated seawater N$_2$O concentration, calculated for in situ temperature and salinity using the solubility data of **Weiss and Price (1980)**. $K$ is the gas transfer velocity, which in the absence of direct measurements can be expressed as a

function of the wind speed and the Schmidt Number ($Sc$). For this study we sourced daily average wind speed from the Cape Sorrel Weather Station at the northern end of Macquarie Harbour (http://www.bom.gov.au/climate/data/index.shtml station ID 097000; see **Figure 1** for station location). $K$ was estimated using relationships in **Raymond and Cole** (**2001**). Fluxes at Macquarie Harbour's endmember stations used $K$ values that account for additional forcings like bottom sheer (*see* **Raymond and Cole 2001**; **Zappa et**

**al., 2003**; **Abril and Borges 2004, Beaulieu et al., 2012; Rosentreter et al., 2021**). Deeper stations in the harbour's main body (*i.e.* KR1, C10, C08, WH2) have surface layers which are separated from the seabed by more than 10 meters. A wind-based K$_{600}$ estimator was used to estimate air-sea flux in those locations (*see* **Raymond and Cole 2001**). Atmospheric N$_2$O for this estimation was sourced from monthly mean baseline greenhouse gas mole fractions measured at the Kennaook / Cape Grim Baseline Air Pollution Station, located in

north west Tasmania. This station measures atmospheric N$_2$O using a gas chromatograph (GC) equipped with an ECD (https://www.csiro.au/en/research/natural-environment/atmosphere/latest-greenhouse-gas-data). N$_2$O saturation (in %) were computed as N$_2$O saturation = 100 * C$_{obs}$ *C$_{eq}^{-1}$.

### 2.5.3    Estimation of Diapycnal N$_2$O Flux

N$_2$O diapycnal fluxes ($F_{dia}$ ; **Equation 3**) from basin waters (sample depths of 20m or 25m) to the harbour's surface lens (sample depths of 2m) were estimated as:



**Equation 3**

$$F_{dia} \ = \ K\rho \, \frac{d[N_2O]}{dz}$$

Where $z$ is depth. Diapycnal diffusivity ($K_\rho$; **Equation 4**) was computed with the local buoyancy frequency ($N^2$),
$\Gamma$ set to 0.2 (Osborn 1980), and $\varepsilon$ the dissipation rate of turbulent kinetic energy assumed to be on the upper end
of values for the mixing zone of stratified systems 1 x 10$^{-5}$ (**Arneborg *et al.,* 2004; Mickett *et al.,* 2004; Fer *et
al.,* 2006**).

**Equation 4**

$$K_\rho \ = \ \Gamma \frac{\varepsilon}{N^2}$$

### 2.6    Data Analysis

The relationships between $N_2O$ saturation and water quality parameters such as DO concentration, salinity,
temperature, nitrate, and ammonium concentrations were analysed using Pearson correlation. The effects of
season and depth on $N_2O$ saturation at each sampling station was tested using a 2-way ANOVA. The relationship
between rainfall / river flow metrics from the Gordon River and surface water $N_2O$ saturation / $N_2O$ air/sea flux
at each station was analysed using Pearson correlation.



### 3. Results

#### 3.1 Rainfall and River Loading

Twenty-day rainfall accumulation ranged from a low of 117 mm in July 2022 to a high of 139 mm in April 2023 with no detectable seasonal differences (*see* **Figure 2A**). Average daily rainfall was similar across all months and ranged from 5.12 (± 2.57) mm in July 2022 to 5.79 (± 3.03) mm in October 2022 (*see* **Figure 2B**). As observed with accumulation metrics no seasonal differences were detected.

Estimated flow at the Gordon River mouth and GAD stream gauge were greater in July and October 2022 than February and April 2023 (**Figure 2C**). At the GAD stream gauge, average flows were observed to decrease during the study period. Greatest flow was observed in winter (July 2022) at 107.6 (± 15.9) $m^3$ $s^{-1}$ and lowest in autumn (April 2023) at 30.5 (± 2.2) $m^3$ $s^{-1}$ (**Figure 2D**).

Estimated $NO_3^-$ and TAN loading varied with $NO_3^-$ loads of 1.69 tonnes $day^{-1}$ observed in July 2022, dipping to 0.31 tonnes $day^{-1}$ in October 2022, and then rising again to 1.77 and 2.77 tonnes $day^{-1}$ in February and April 2023 (**Figure 2E**). TAN loading mirrored this pattern with peaks occurring in October 2022 and February 2023 and lows occurring in July 2022 and April 2023. Patterns in $N_2O$ loading from the Gordon River were similar to those observed for $NO_3^-$ (**Figure 2F**).

#### 3.2 Water Column Physicochemical Profiles

DO profiles at the stations located within the main body of the harbour show a well oxygenated surface layer that rapidly attenuates with depth (**Figure 3A**) through the halocline (**Figure 3B**). There is a prominent riverine surface lens in the main harbour extending to depths of up to 8m depending on sampling period and location

within the estuary. Salinity in the surface waters was lower in July and October 2022 (6 to 13) than February and April 2023 (greater than 20). Below the halocline salinity ranged from approx. 28 to 32.

The DO gradient between the surface and subhalocline waters was steeper in October relative to July 2022 with October 2022 DO concentrations approaching single digits (3.1 µM) at station WH2, nearest the Gordon River

mouth (*see* **Figure 1**). In general, the subhalocline concentrations of DO were lower with proximity to the Gordon River mouth. The temperature of the freshwater surface layer ranged from about 9 ℃ to 19 ℃, but showed little variation below the halocline where temperature ranged between 13 ℃ to 16 ℃ (**Figure 3C**).





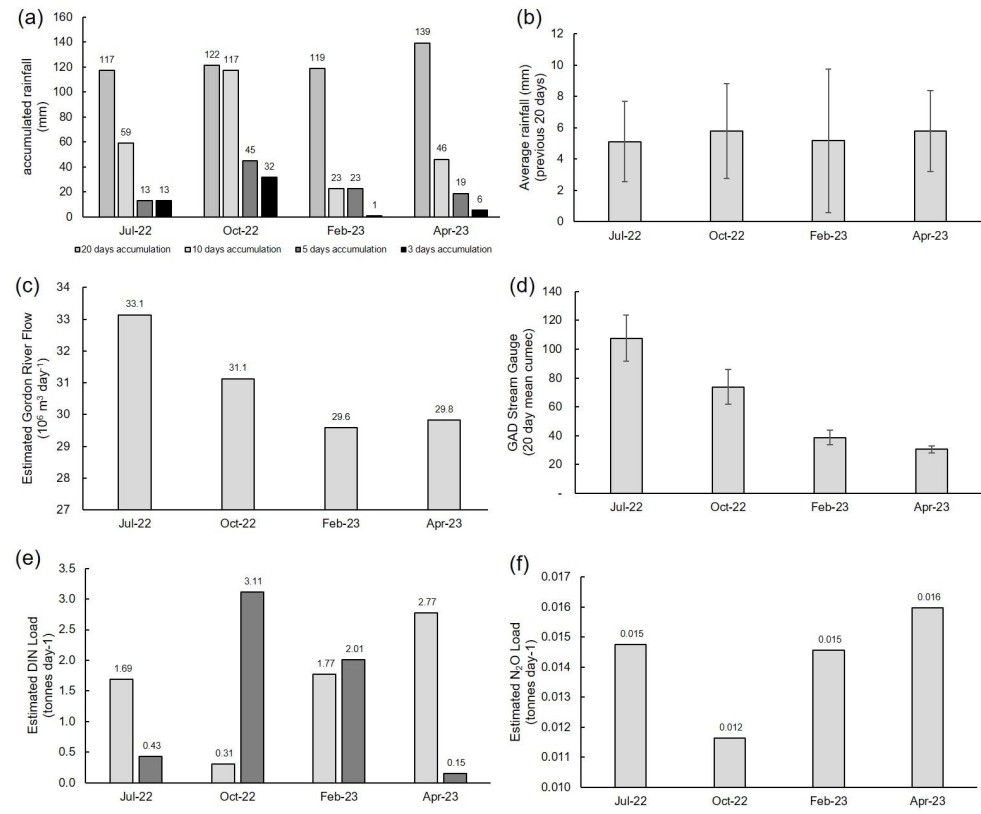

**Figure 2: Rainfall and estimated Gordon River loading estimates for each sampling event. A) accumulated rainfall (mm) 10, 5, and 3 days prior to each sampling event; B) average (mean) daily rainfall over a 20 day period prior to each sampling event; C) estimated Gordon River Flow into the harbour in millions of m³ day⁻¹; D) daily mean flow (m³ sec⁻¹) over previous 20 days prior to sampling (± standard error) at the Gordon Above Denison Stream Gauge; E) estimated nitrate and ammonium loads entering the harbour from the Gordon River; F) estimated N₂O load (tonnes day⁻¹) entering the harbour from the Gordon River.**

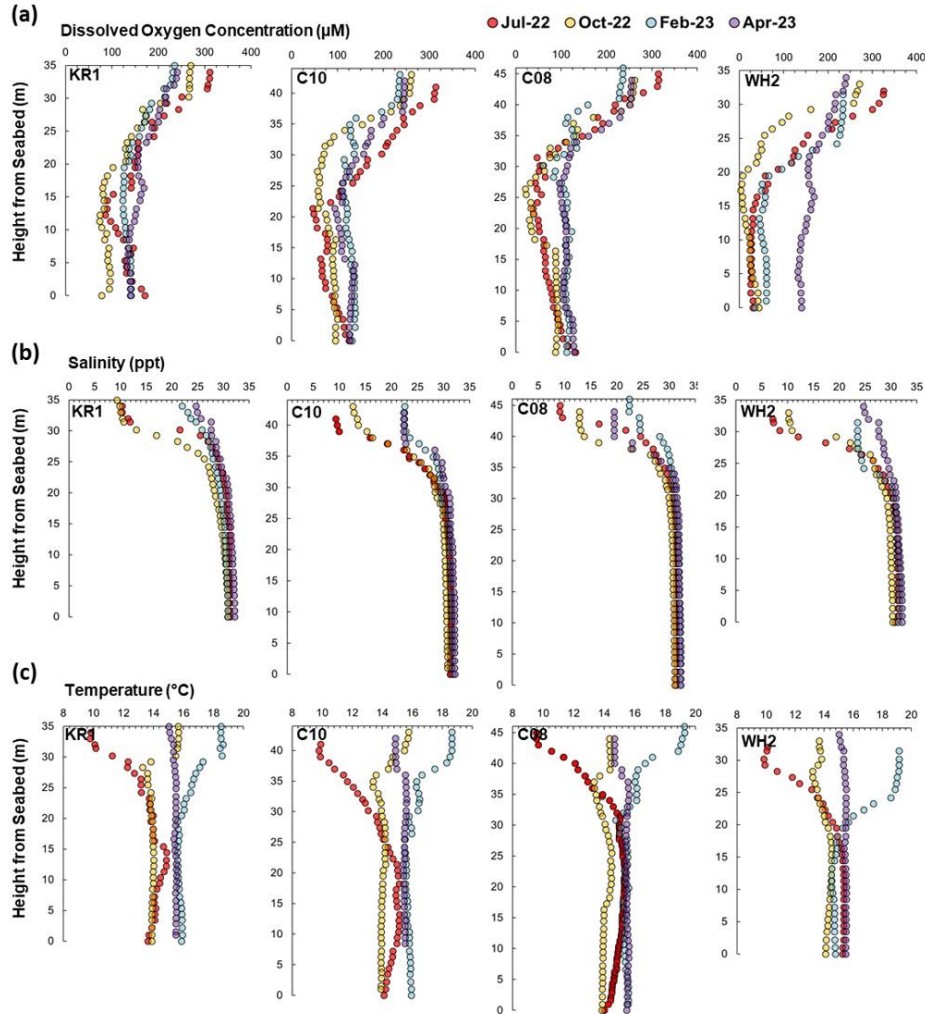

**Figure 3: Dissolved oxygen (µM) (Row A), salinity (Row B), and temperature (°C) (Row C) profiles (referencing height from seabed) collected at stations KR1, C10, C08, and WH2 in July 2022 (red dots), October 2022 (yellow dots), February 2023 (blue dots), and April 2023 (purple dots). Measurements were made every 1 meter.**

### 3.3    N$_2$O Distribution

At each harbour station, depth and season (and their interaction) significantly impacted N$_2$O saturation (two-way ANOVA, α = 0.05, *degree of freedom (d.f.)* = 59). At 2 m, N$_2$O saturation was observed to be below 100% at all stations in July 2022 (**Figure 4**) and at stations KR1, C10, and C08 in October 2022. In February and April 2023 N$_2$O saturation in the harbour was above 100% through the water column except in KR1 surface waters. The

maximum N$_2$O concentrations were observed in the subhalocline. Among the subhalocline observations the maximum N$_2$O concentrations (reaching over 170%) were observed at the base of the Hells Gates sill at station C10 in October 2022.



All endmember $N_2O$ concentrations were undersaturated in July 2022. In October, stations KR1 and HG3 were observed to be approx. 100% saturated but $N_2O$ at station GR1 was undersaturated. In February and April 2023

$N_2O$ concentrations were supersaturated at all endmember stations. There were statistically significant linear correlations between $N_2O$ saturation and salinity (r = 0.494; p = 5.5 x $10^{-7}$, n = 92), temperature (r = 0.391; p = 1.2 x $10^{-4}$, $d.f.$ = 90), DO concentration (r = -0.563; p = 5.2 x $10^{-9}$, $d.f.$ = 90), and nitrate concentration (r = 0.559; p = 6.9 x $10^{-9}$, $d.f.$ = 90) in the harbour stations (**Figure 5**). The correlation between $N_2O$ saturation and the TAN concentration however was not statistically significant (r = 0.174; p = 0.31, $d.f.$ = 34).





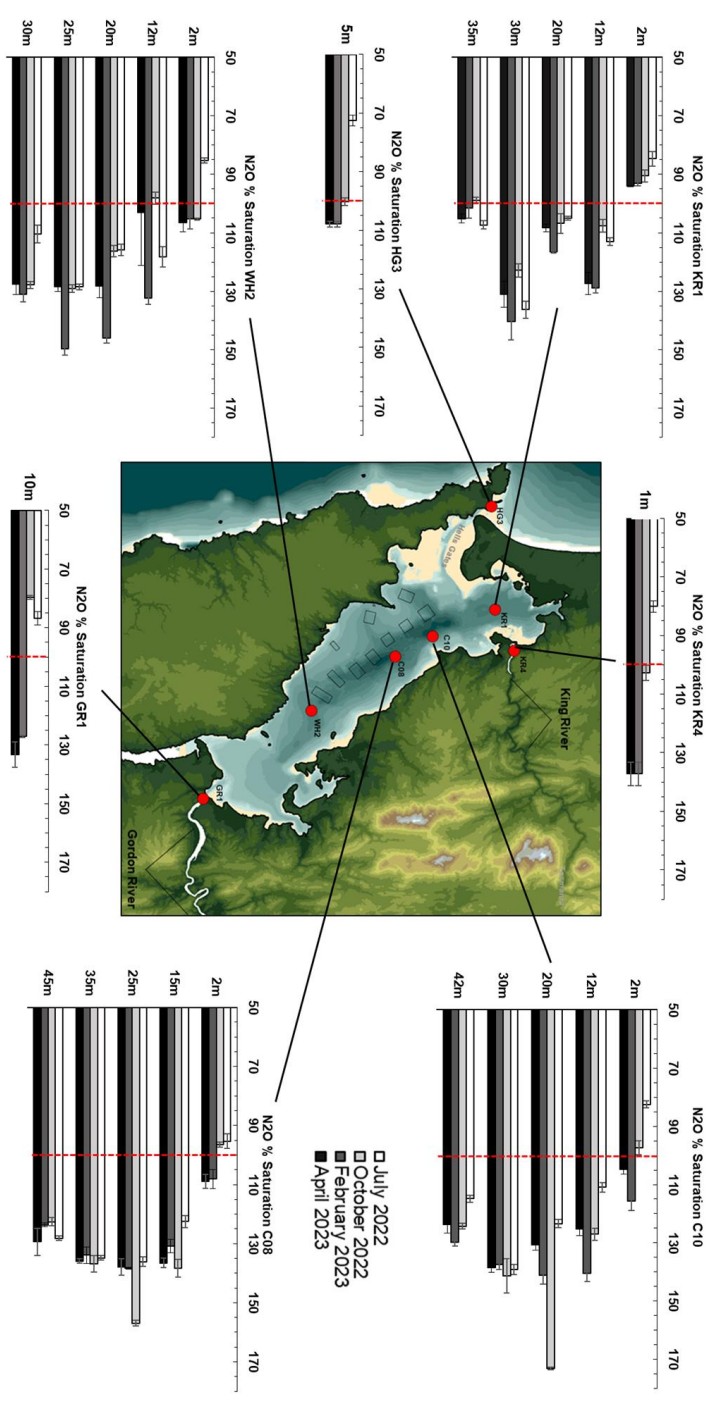

**Figure 4: Mean (± standard error) N₂O % saturation observed at each sampling station, with depth, and across seasons. Note that a red dashed line indicating 100% at the time of sampling has been placed on each panel for reference.**




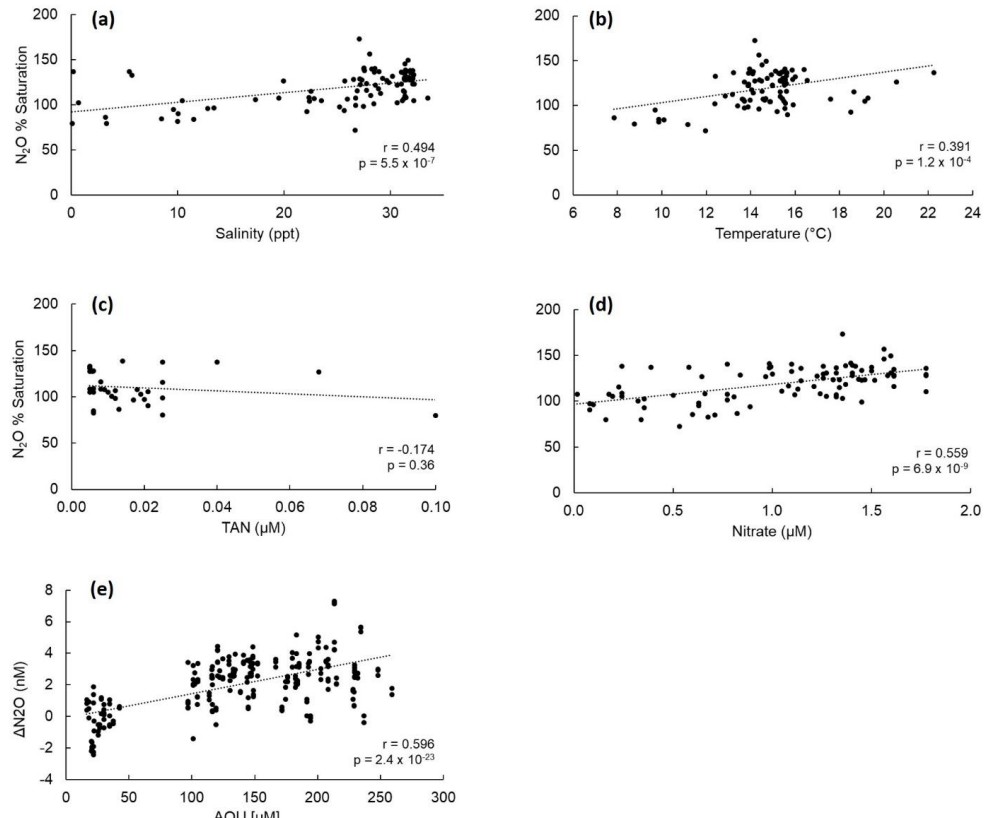

**Figure 5: Correlation between N₂O % Saturation observed across the harbour and A) Salinity, B) Temperature, C) Total Ammoniacal Nitrogen (TAN) concentration, D) Nitrate concentration. The correlation between AOU [μM] and ΔN₂O [nM] is shown in panel  E. Pearson correlation coefficients (r) and their associated p value are shown in each panel.**

### 3.4    N₂O Air/Sea and Diapycnal Fluxes

Atmospheric N₂O  mole fractions measured at Kinnaook / Cape Grim Air Pollution Station (*see* **Figure 1**) were observed to increase from 334.7 ppb in July 2022 to 335.9 ppb in February 2023 (*see* Error! Reference source not found.). The April 2023 atmospheric N₂O mole fraction was slightly lower than that observed in February 2023 at 335.6 ppb.

Estimated N₂O air/sea flux in the main harbour stations (KR1, C10, C08, WH2) was observed to range from -12.88 (± 0.88) μmol N₂O m$^{-2}$ day$^{-1}$ at C10 in July 2022 (negative sign indicates absorption of N₂O into the surface waters from the atmosphere) to 7.31 (± 0.88) μmol N₂O m$^{-2}$ day$^{-1}$ at the same station in February 2023 (using the "High" $K_{600}$ estimator from **Raymond and Cole (2001)**; *see* **Table 2**).

Station KR1 was always observed to be a net sink for atmospheric N₂O, and every non-endmember station was an estimated sink in July 2022. Near the head of the system, station WH2 was observed to be a net source of



N$_2$O to the atmosphere in October 2022, February 2023, and April 2023, as were stations C10 and C08 (positioned above the deepest basins) in February 2023 and April 2023.

Estimated diapycnal fluxes using local buoyancy frequencies showed a consistent upwards movement of N$_2$O from the subhalocline to surface layers with the smallest fluxes observed in July 2022 (49 nmol N$_2$O m$^{-2}$ day$^{-1}$ at 320  C08) and largest fluxes observed in October 2022 (up to 1308 nmol N$_2$O m$^{-2}$ day$^{-1}$ at WH2) and February 2023 (up to 1200 nmol N$_2$O m$^{-2}$ day$^{-1}$ at C10)  (**Table 3**). Patterns in the size of the diapycnal flux generally reflected the patterns of N$_2$O % saturation with the largest fluxes occurring in October 2022 during the periods of greatest N$_2$O % saturation. Overall the magnitude of the estimated diapycnal fluxes was smaller than estimated air/sea fluxes ( smaller) however in February the fluxes were of similar magnitudes.



**Table 2: Estimated sea-to-air N₂O flux (mean µMol N₂O m⁻² day⁻¹ ± standard error) of the main harbour stations using calculations presented in Bange et al. (2019) and Zhang et al. (2020) and a range of k₆₀₀ estimators from Raymond and Cole (2001). Low, Mid, and High represent different estimators of k₆₀₀ presented in Raymond and Cole (2001). Positive values indicate the flux of N₂O from the harbour water to the atmosphere. Negative values (shown in with bold text) indicate flux of N₂O from the atmosphere into the harbour water. Estimated Gordon River Flow and Mean (20 day) Gordon Above Dennison (GAD)Stream Gauge are also shown for each month as well as the Pearson Correlation and associated p-values between flow metrics, rainfall, and sea-to-air flux (and surface water % saturation).**

| Station | $K_{600}$ Est. | Jul 2022 $\mu mol\ N_2O\ m^{-2}\ day^{-1}$ | Oct 2022 $\mu mol\ N_2O\ m^{-2}\ day^{-1}$ | Feb 2023 $\mu mol\ N_2O\ m^{-2}\ day^{-1}$ | Apr 2023 $\mu mol\ N_2O\ m^{-2}\ day^{-1}$ | Gordon Flow vs Surface Flux | GAD Flow vs Surface Flux | GAD Flow vs % N₂O Sat. | Rainfall vs Surface Flux |
|---|---|---|---|---|---|---|---|---|---|
| KR1 | High: Mid: Low: | **-11.07 ± 1.84** **-08.45 ± 1.41** **-04.69 ± 0.78** | **-04.01 ± 0.86** **-03.19 ± 0.69** **-01.93 ± 0.42** | **-03.30 ± 0.49** **-02.55 ± 0.38** **-01.46 ± 0.22** | **-03.17 ± 0.16** **-02.44 ± 0.12** **-01.38 ± 0.07** | r = -0.8316 p = 7.5 x 10⁻⁴ | r = -0.8624 p = 3.1 x 10⁻⁴ | r = -0.8726 p = 2.1 x 10⁻⁴ | r = 0.5577 p = 0.060 |
| C10 | High: Mid: Low: | **-12.88 ± 0.88** **-09.83 ± 0.67** **-05.46 ± 0.37** | **-01.21 ± 1.07** **-00.96 ± 0.85** **-00.58 ± 0.51** | 07.31 ± 1.57 05.65 ± 1.22 03.22 ± 0.69 | 02.60 ± 0.85 02.00 ± 0.66 01.13 ± 0.37 | r = -0.8298 p = 8.4 x 10⁻⁴ | r = -0.9091 p = 4.2 x 10⁻⁵ | r = -0.8795 p = 1.6 x 10⁻⁴ | r = 0.2751 p = 0.387 |
| C08 | High: Mid: Low: | **-03.50 ± 1.82** **-02.67 ± 1.39** **-01.49 ± 0.77** | **-01.69 ± 0.38** **-01.34 ± 0.29** **- 0.81 ± 0.18** | 04.08 ± 1.07 03.15 ± 0.83 01.80 ± 0.47 | 04.57 ± 1.79 03.52 ± 1.38 01.98 ± 0.78 | r = -0.8547 p = 3.97 x 10⁻⁴ | r = -0.8804 p = 1.6 x 10⁻⁴ | r = -0.8447 p = 5.4 x 10⁻⁴ | r = 0.1846 p = 0.566 |
| WH2 | High: Mid: Low: | **-10.88 ± 0.68** **-08.30 ± 0.52** **-04.61 ± 0.29** | 02.63 ± 0.17 02.09 ± 0.14 01.26 ± 0.08 | 02.40 ± 1.56 01.85 ± 1.20 01.06 ± 0.69 | 03.50 ± 1.72 02.69 ± 1.33 01.52 ± 0.75 | r = -0.8071 p = 1.51 x 10⁻³ | r = -0.8269 p = 9.1 x 10⁻⁴ | r = -0.8077 p = 1.5 x 10⁻³ | r = 0.6316 p = 0.028 |
| Gordon River Flow (m³ sec⁻¹) | | 383.6 ± 38.9 | 360.3 ± 54.1 | 342.6 ± 74.6 | 324.3 ± 26.6 | - | - | - | - |
| GAD Flow (m³ sec⁻¹) | | 107.6 ± 15.9 | 73.7 ± 12.1 | 38.8 ± 5.1 | 30.5 ± 2.2 | - | - | - | - |

**Table 3: Estimated diapycnal N₂O flux (nmol N₂O m⁻² day⁻¹) from 20 m to 2 m within the main harbour stations Positive values indicate the flux of N₂O from the basin water (20 m) to the surface lens (2m).**

| Station | July 2022 $nmol\ N_2O\ m^{-2}\ day^{-1}$ | October 2022 $nmol\ N_2O\ m^{-2}\ day^{-1}$ | February 2023 $nmol\ N_2O\ m^{-2}\ day^{-1}$ | April 2023 $nmol\ N_2O\ m^{-2}\ day^{-1}$ |
|---|---|---|---|---|
| KR1 | 80 | 282 | 992 | 395 |
| C10 | 140 | 1200 | 1040 | 454 |
| C08 | 49 | 782 | 778 | 348 |
| WH2 | 117 | 125 | 1308 | 240 |



**4. Discussion**

Our study is the first to report on $N_2O$ distribution and air/sea flux from an Australasian fjord-like estuary. We set out to investigate how $N_2O$ concentrations varied along horizontal and depth gradients; how $N_2O$ concentrations and estimated surface water emissions vary seasonally; how $N_2O$ concentrations vary with freshwater inputs; and whether the relationship between AOU and $\Delta N_2O$ could help clarify the primary mechanism for $N_2O$ generation in this system.

We used surface water observations, local wind speed (from Cape Sorell weather station) and atmospheric $N_2O$ mole fractions (from Cape Grimm; **Figure 1**) to estimate $N_2O$ air/sea flux (based on **Zhang et al., (2010)** and **Bange et al., (2019)**) and found that Macquarie Harbour functions as both a sink and a source of $N_2O$. Most harbour stations were estimated to be a sink for $N_2O$ in July and October 2022 (when river flow was greater) and a source in February and April 2023 (during low river flow periods; *see* **Figure 6** and **Table 2**). Pearson correlations show that when freshwater flow is high $N_2O$ air/sea flux is negative (indicating uptake from the atmosphere) and when freshwater flow is low $N_2O$ air/sea flux is positive (**Table 2**). Our observations highlight that freshwater flow is a key driver of $N_2O$ emissions in this estuary. In addition, Gordon River flow is heavily influenced by hydroelectric dam release (up to ~28% of the flow in July 2023). Rainfall in the catchment area may offset the effects of dam release, but our observations did not capture this as rainfall itself was not significantly correlated with $N_2O$ concentrations or air/sea flux.

The river endmember concentrations of $N_2O$ were often observed to be undersaturated, as observed in the South Platte River Basin, USA, **McMahon and Dennehy 1999**; Neuse River Estuary, USA, **Stow et al., 2005**; headwater streams, Ontario, Canada, **Baulch et al., 2011**; Upper Mara River Basin, Kenya, **Mwanke et al., 2019.** Our observations of endmember $N_2O$ concentrations were similar to the lower end of the concentrations reported in **McMahon and Dennehy (1999)** (approx. 80% saturation), but not as low as those reported Jackson Creek, Ontario, Canada in **Baulch et al., (2011)** were some observations reached <20% saturation. $N_2O$ undersaturation in those systems was attributed to complete denitrification (use of $N_2O$ as a terminal electron acceptor by denitrifies) in streams with high DOC loads, low DO, low $NO_3^-$ concentrations. It should also be noted that up to 28% of the estimated Gordon River flow was found to be associated with flow through the Gordon Above Dennison stream gauge (a proxy for hydroelectric dam/reservoir release to the Gordon River). Boreal reservoirs have been shown to be net sinks of atmospheric $N_2O$ (**Hendzel et al., 2005**) which was attributed to increased $N_2O$ demand to drive complete denitrification. There is good reason to believe that $N_2O$ may be scavenged in the Gordon and King Rivers as well because they do often have high DOC concentrations, high water column DO demand (**Maxey et al., 2020**), and low DO concentrations in near the stream bed (**Maxey et al., 2022**).

Below the estuary's predominately freshwater surface lens, the fjord-like morphology drives suboxic conditions like those observed in the subhalocline waters at station WH2 in October 2022 (*see* **Figure 3; Hartstein et al., 2019; Maxey et al., 2020, 2022**). While these conditions do not always persist, DO concentrations below 31 µM have been observed to occur more than 30% of the time up estuary, specifically at station WH2 (**Maxey et al., 2022**). In layers of the harbour where DO concentrations were lowest (subhalocline layers) we observed the





maximum $N_2O$ concentrations (**Figure 4**). Subhalocline $N_2O$ saturation was observed to generally range from

approx. 110% to 170% with the highest values observed within the deeper basins near the foot of the sill

(stations C10 and C08).

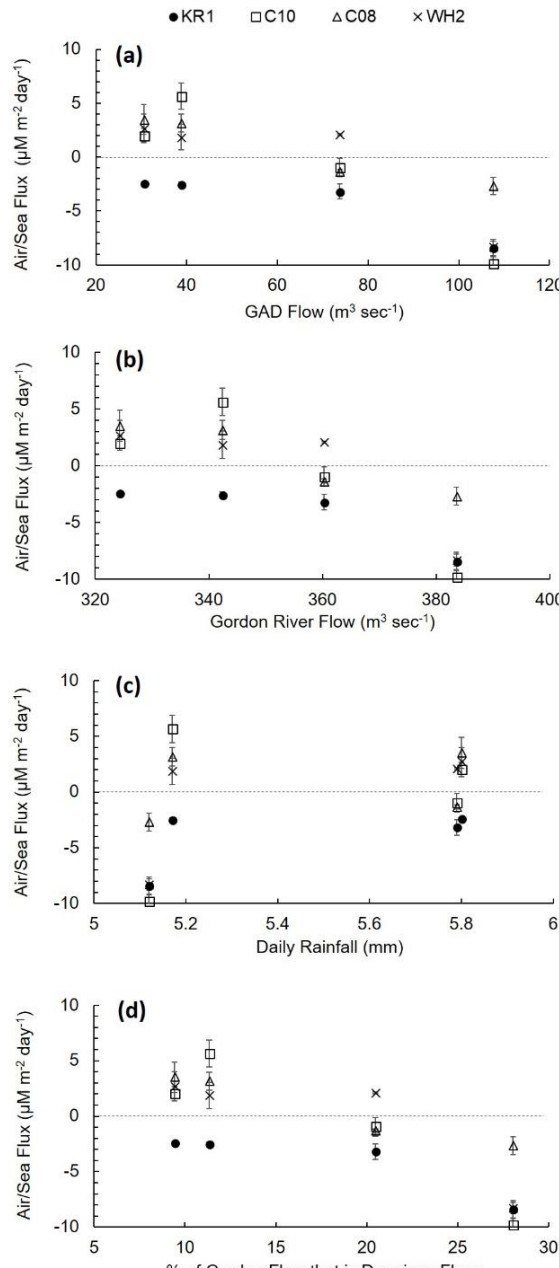

384

**Figure 6: Mean Air/Sea Flux (µM m-2 day-1) versus A) Gordon above Dennison River flow (m3 day-1), B) estimated Gordon River flow (m3 day-1), C) daily rainfall (mm) (20 day mean), and D) % of estimated Gordon River flow this is accounted for by the Gordon above Dennison River gauge (proxy for hydroelectric dam release). Error bars indicate ± 1 standard error.**



In the harbour's subhalocline layer there is not enough light to support photosynthesis (**Hartstein *et al.,* 2019**;
**Maxey *et al.,* 2017, 2020, and 2022**) and thus the main source of oxygen is advection from marine intrusions.
$N_2O$ producing microbes have been observed to populate this layer of the harbour (*see* **Da Silva *et al.,* 2021 and**
**2022**) and our observations of supersaturated $N_2O$ in these layers show that those microbes are active. The linear
relationship between AOU and $\Delta N_2O$ (slope = 0.0154; r = 0.596; p = 2.4 x $10^{-23}$; **Figure 5C**) indicates that $N_2O$
production occurs primarily through the ammonia oxidation (nitrification) pathway (**Yoshinari, 1976; Walter *et***
***al.,* 2004; Brase *et al.,* 2017**). Our observations are on the lower end of the range reported nitrification slopes
(see **Suntharalingam and Sarmiento, 2000; Brase *et al.,* 2017**) indicating a low yield of $N_2O$ per mole $O_2$
consumed. These more modest yields are likely an artefact of mixing and loss dynamics such as basin water DO
recharges from marine intrusions, and loss to aerobic respiration and loss to the atmosphere. This suggests that
some portion of subhalocline pelagic oxygen demand in the harbour can be attributed to nitrifying microbes
(albeit at a much lower rate compared to aerobic respiration). **Ji *et al.,* (2020)** also observed similar relationships
in the Saanich Inlet, a seasonally anoxic fjord-like estuary in British Columbia, but in that system anoxic
conditions are more persistent (**Bourbonnais *et al.,* 2013; Manning *et al.,* 2010**) compared to Macquarie
Harbour (**Maxey *et al.,* 2022**). Deep-water renewal / marine intrusions have been hypothesized to stimulate $N_2O$
production in the Saanich Inlet (**Capelle *et al.,* 2018; Michiles *et al.,* 2019**; **Ji *et al.,* 2020**), and Baltic Sea
(**Walter *et al.,* 2006**) and may also be stimulating it in Macquarie Harbour as well. In the Baltic Sea, **Walter *et***
***al.* (2006)** and **Myllykangas *et al.* (2017)** observed enhanced $N_2O$ production in areas receiving significant
marine intrusions. Positive correlations between AOU and $\Delta N_2O$ observed in western Baltic Sea waters (**Walter**
***et al.,* 2006**) along with mean (11-year; 2006-2017) seasonal variations in DO and $N_2O$ observed through the
water column at the Boknis Eck Time-Series Station (Eckernförde Bay, Southwest Baltic Sea) indicate a tight
coupling between DO supply and $N_2O$ production (presumably by nitrification) / consumption (presumably by
denitrification) pathways in that area (**Ma *et al.,* 2019**). The reintroduction of marine water on the upstream side
of a dam in the Nakong River, South Korea was found to affect bottom water trapping (stagnation), DO
conditions, N process rates, process specific gene abundances, and subsequently the fate of N in that system
(**Huang *et al.,* 2024**). Marine intrusions primarily refresh the DO supply adjacent to the sill in Macquarie
Harbour (near station C10) and since we also observed a positive correlation between AOU and $\Delta N_2O$ they offer
a possible explanation for the higher subhalocline $N_2O$ concentrations observed there.

We conceptualize that during periods of high river flow, the surface water lens thickens and transports water
undersaturated with $N_2O$ quickly across the harbour surface and out of Hells Gates inlet. Some $N_2O$ from the
oversaturated subhalocline water is entrained in the surface lens (diapycnal flux) and transported out of the
system in its dissolved form. During periods of low river flow, the surface lens is thinner and residence times
longer (**Andrewartha and Wild-Allen 2017; Maxey *et al.,* 2022**). $N_2O$ from the oversaturated subhalocline
water then diffuses through the surface layer and escapes into the atmosphere in its gaseous form (**Figure 7**). Our
estimates of diapycnal flux indicate that the mass transport from subhalocline waters is smaller (~2x smaller)
than the air/sea flux supporting this idea. This conceptual model suggests that the harbour surface lens functions
as a sink for both atmospheric $N_2O$ and $N_2O$ generated in the subhalocline layer during high flow periods
(**Figure 7**).





Previous work in Australian estuaries with pristine catchments (like Macquarie Harbour) has shown that many
tend to function as a sink for atmospheric $N_2O$ (**Maher *et al.,* 2016; Wells *et al.,* 2018**). Our study adds the
caveat that source sink dynamics may also depend on factors controlling river flow in deeper stratified systems.
Despite the advancements made to date, many of the deeper estuaries in Chile, Australia and New Zealand are
lacking descriptions of $N_2O$ source sink dynamics (*e.g.* Bathurst Harbour, Tasmania; fjords of South Island New
Zealand; estuaries on Stewart Island New Zealand). Given that these systems have relatively pristine catchments
they offer an opportunity to better understand natural fjord-like estuarine responses to the climate drivers of $N_2O$
dynamics. Mesoscale climate oscillations (*i.e.* the Southern Annular Mode; SAM; North Atlantic Oscillation;
NAO) have been shown to affect rainfall, river flow, and dissolved oxygen concentrations in this and other fjord-
like estuaries (**Maxey *et al.,* 2022**; **Austin and Inall, 2002**). In Western Tasmania, SAM in its positive phase
results in increased orographic rainfall and a greater propensity for higher river flow, possibly tilting the source
and sink balance to net $N_2O$ uptake during these periods.

Climate change predictions for Tasmania's West Coast (which includes the Macquarie Harbour catchment)
indicate that the region will experience a more extreme precipitation regime with increased winter precipitation
and decreased summer precipitation (**Grose *et al.,* 2010; Bennett *et al.,* 2010**). If these future predictions result
in more extreme seasonality in Gordon River flow, then the harbour may respond in kind with a larger variation
in $N_2O$ source and sink dynamics *i.e.* larger $N_2O$ sink in winter and $N_2O$ source in summer. However, given that
the river flow is somewhat regulated by the hydroelectric dam, our study suggests that flow regulation has the
potential to augment harbour $N_2O$ emissions. Releasing water during extreme low rainfall periods might allow
$N_2O$ slowly accumulating in subhalocline waters to be released in the exported surface lens.

It is well established that fjord and fjord-like estuaries are important sites of C burial (**Smith *et al.,* 2015;**
**Bianchi *et al.,* 2018, 2020**). This study supports the idea that they can also be important sites of $N_2O$
sequestration. Macquarie Harbour air/sea flux estimates are similar in magnitude to observations made in other
stratified estuaries and enclosed seas such as the Reloncaví Estuary, Chile (**Yevenes *et al.,* 2017**) and
Eckernförde Bay, Germany (**Ma *et al.,* 2019**) (**Table A1**). Macquarie Harbour, however, was observed to have
lower fluxes of $N_2O$ into the atmosphere than other river dominated, but not fjord-like, estuaries (Elbe River,
Germany; **Schulz *et al.,* 2023**) including those on the Australian mainland's east coast (**Wells *et al.,* 2018**).

Fjord and fjord-like estuaries are defined by their strong stratification and sensitivity to freshwater inputs. With
climate change, rainfall patterns are expected to become more extreme and thus alter the river flow, and
subsequently $N_2O$ source sink dynamics in these systems on a global scale. In systems that are expected to
experience increasingly drier conditions they may shift from net sinks of $N_2O$ to sources, and further perpetuate
the accumulation of $N_2O$ in the atmosphere.
**5.  Conclusions**
In summary, river flow, and specifically river flow driven by hydroelectric dam release, significantly affects both
surface water $N_2O$ concentrations and air/sea flux in Macquarie Harbour. Importantly, when river flow is low



most of the harbour emits $N_2O$ to the atmosphere. When river flow is high most of the harbour removes $N_2O$
from the atmosphere and intercepts the diapycnal flux exporting $N_2O$ to the ocean in its dissolved form.

$N_2O$ is continually supersaturated below the halocline and the relationship between AOU and $\Delta N_2O$ indicate that
the main $N_2O$ generation process is nitrification. Climate change is predicted to result in wetter winter / drier
summers for the Tasmanian West Coast, which may result in augmented $N_2O$ air/sea fluxes.

These represent the first descriptions of $N_2O$ spatiotemporal distribution, estimated air/sea flux, estimated
diapycnal flux, and $N_2O$ production pathways in this system.

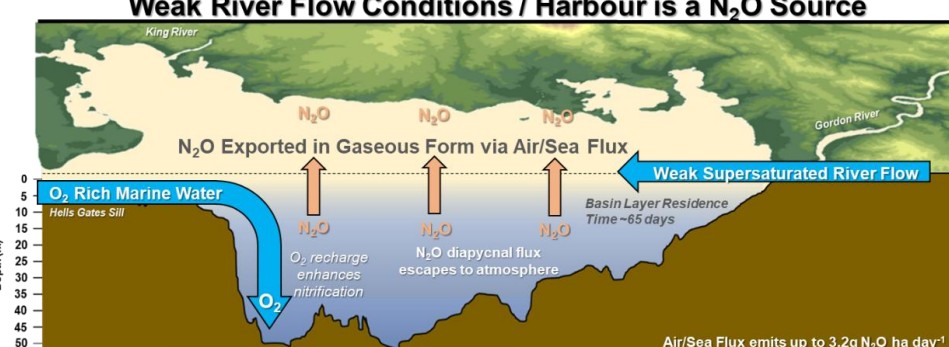


**Figure 7: Conceptual model of Macquarie Harbour's $N_2O$ dynamics. The top diagram depicts the capture of $N_2O$**
**generated in the subhalocline during strong river flow conditions. Here $N_2O$ is exported from the harbour in its**
**dissolved form via undersaturated surface flows from the harbour to the ocean. The bottom diagram depicts the**
**efflux of $N_2O$ from the harbour surface during low flow conditions. Note that during these conditions the surface**
**flows are weak and generally supersaturated with $N_2O$ permitting its escape in gaseous form to the atmosphere.**



## 6. Appendix

**Table A1: N₂O fluxes and observed ranges of mean (± standard deviation) N₂O concentration / saturation from both fjord-like / river dominated estuaries around the globe and estuaries in Australia.**

| Location | System Type | Measurement Depth Range | Mean Sea-to-Air N₂O flux uMol N₂O m⁻² day⁻¹ | Min and Max Sea-to-Air N₂O flux uMol N₂O m⁻² day⁻¹ | Mean N₂O Concentration (and Saturation) nM N₂O (and %) | Min and Max N₂O Concentration (and Saturation) nM N₂O (and %) | Reference |
|---|---|---|---|---|---|---|---|
| Macquarie Harbour, Western Tasmania, Australia | Fjord-like Estuary | 2m to 45m | -09.83 ± 0.67 to 05.65 ± 1.22 | -10.82 to 7.73 | 11.7 ± 1.6 (121.8 ± 17.8) | 7.87 to 17.12 (81 to 174) | *This Study* |
| Reloncaví Estuary, Chile | Fjord-like Estuary | 0m to 5m | 0.86 ± 2.28 | -1.58 to 5.60 | 11.8 ± 1.70 (111 ± 18.3) | 8.34 to 14.5 (80 to 140) | Yevenes *et al.*, 2017 |
| Reloncaví Estuary, Chile | Fjord-like Estuary | 10m to 200m | - | - | 14.5 ± 1.73 (145 ± 17.7) | 10.5 to 17.0 (11 to 170) | Yevenes *et al.*, 2017 |
| Chiloé Interior Sea, Chile | Fjord-like Estuary | 0m to 200m | 1.08 ± 1.41 | -0.18 to 3.19 | 12.6 ± 2.36 (121 ± 17.5) | 8.81 to 21.1 (87 to 160) | Yevenes *et al.*, 2017 |
| Europa Sound, Magellanic Region, Chile | Fjord-like Estuary | 1m to 10m | -15.22 to -0.81 | - | 11.9 ± 5.7 to 12.7 ± 1.0 | - | Farías *et al.*, 2018 |
| Concepción Channel, Magellanic Region, Chile | Fjord-like Estuary | 1m to 150m | 0.69 to 7.70 | - | 13.6 ± 1.1 to 17.0 ± 0.02 | - | Farías *et al.*, 2018 |
| Sarmiento Channel, Magellanic Region, Chile | Fjord-like Estuary | 1m to 10m | 2.07 to 12.53 | - | 13.1 ± 0.1 to 16.5 ± 0.3 | - | Farías *et al.*, 2018 |
| Estero Peel, Magellanic Region, Chile | Fjord-like Estuary | 1m to 10m | 0.11 to 2.01 | - | 13.1 ± 0.2 to 13.5 ± 0.5 | - | Farías *et al.*, 2018 |
| Estero Calvo, Magellanic Region, Chile | Fjord-like Estuary | 1m to 10m | 0.04 | - | 13.9 ± 0.8 | - | Farías *et al.*, 2018 |
| Estero Amalia, Magellanic Region, Chile | Fjord-like Estuary | 1m to 100m | -0.08 | - | 14.2 ± 1.7 | - | Farías *et al.*, 2018 |
| Estero las Montañas, Magellanic Region, Chile | Fjord-like Estuary | 1m to 10m | -2.95 | - | 9.69 ± 1.6 | - | Farías *et al.*, 2018 |
| Smyth Channel, Magellanic Region, Chile | Fjord-like Estuary | 1m to 300m | 1.07 to 11.2 | - | 14.3 ± 0.4 to 16.0 ± 0.5 | - | Farías *et al.*, 2018 |
| Última Esperanza Sound, Magellanic Region, Chile | Fjord-like Estuary | 1m to 10m | -3.7 to 10.4 | - | 12.1 ± 1.1 to 13.7 ± 0.07 | - | Farías *et al.*, 2018 |
| Almirante Montt Gulf, Magellanic Region, Chile | Fjord-like Estuary | 1m to 150m | 15.6 | - | 21.0 ± 5.7 | - | Farías *et al.*, 2018 |
| Kirke Channel, Magellanic Region, Chile | Fjord-like Estuary | 1m to 10m | 0.12 to 8.19 | - | 13.3 ± 0.1 to 15.4 ± 0.4 | - | Farías *et al.*, 2018 |
| Union Channel, Magellanic Region, Chile | Fjord-like Estuary | 1m to 10m | 22.1 | - | 16.7 ± 0.8 | - | Farías *et al.*, 2018 |
| Union Sound, Magellanic Region, Chile | Fjord-like Estuary | 1m to 10m | 2.86 | - | 14.8 ± 0.8 | - | Farías *et al.*, 2018 |
| Western Magellan Strait, Magellanic Region, Chile | Fjord-like Estuary | 1m to 10m | 143 | - | 15.71 | - | Farías *et al.*, 2018 |
| Eastern Magellan Strait, Magellanic Region, Chile | Fjord-like Estuary | 1m | 36.3 | - | 16.4 | - | Farías *et al.*, 2018 |
| San Gregorio Cape, Magellanic Region, Chile | Fjord-like Estuary | 1m | 24.8 | - | 12.07 | - | Farías *et al.*, 2018 |
| Otway Center Sound, Magellanic Region, Chile | Fjord-like Estuary | 1m | 35.5 | - | 11.4 | - | Farías *et al.*, 2018 |
| Magdalena North Channel, Magellanic Region, Chile | Fjord-like Estuary | 1m | -0.22 | - | 11.4 | - | Farías *et al.*, 2018 |
| Chasco Sound, Magellanic Region, Chile | Fjord-like Estuary | 1m | 6.81 | - | 16.01 | - | Farías *et al.*, 2018 |
| Cockburn West Channel, Magellanic Region, Chile | Fjord-like Estuary | 1m | 6.18 | - | 14.47 | - | Farías *et al.*, 2018 |
| Saanich Inlet, British Columbia, Canada | Fjord-like Estuary | 10m to 200m | 2.3 ± 2.5 to 3.9 ± 2.9 | - | 14.7 | <0.5 to 37.4 | Capelle *et al.*, 2018 |
| Saanich Inlet, British Columbia, Canada | Fjord-like Estuary | Surface to 110m | 11.3 to 20.4 | - | - | - | Cohen 1978 |
| Elbe River Estuary, Germany | Well-Mixed River Dominated Estuary | 1.2m | - | 26.0 ± 23.5 to 100.7 ± 101.2 | - | (161 ± 53.6) to (243 ± 141.6) | Schulz *et al.* 2023 |
| Eckernförde Bay, Boknis Eck Time Series Station, Baltic Sea, Germany | Enclosed Sea | 1m to 25m | 3.5 ± 12.4 | -19.0 to 105.7 | (111 ± 30) | (56 to 314) | Ma *et al.*, 2019 |
| Eckernförde Bay, Boknis Eck Time Series Station, Baltic Sea, Germany | Enclosed Sea | 1m to 25m | - | - | 10 to 17 | - | Walter *et al.*, 2006 |
| Baltic Sea, Germany | Enclosed Sea | 110m | 5 -11 | - | 14 to 1523 | - | Rönner 1983 |



| Location | Type | Depth | | | | | Reference |
|---|---|---|---|---|---|---|---|
| Gotland Basin, Baltic Sea, Germany | Enclosed Sea | 90m | - | - | 13 | 0 to 126 (0 to 450) | Brettar and Rheinheimer 1991 |
| Northwest Shelf, Black Sea | Enclosed Sea | - | 1.6 to 4.4 | - | 6.5 to 8 | - | Amouroux *et al.*, 2002 |
| Deep Basin, Black Sea | Enclosed Sea | 70m | 3.1 to 5.2 | - | 7.5 to 10.2 | - | Amouroux *et al.*, 2002 |
| Cariaco Basin, Venezuela | Coastal Basin | Surface to 400m | - | - | 4.4 to 5.5 | - | Hashimoto *et al.*, 1983 |
| Guadalquivir Estuary, Gulf of Cadiz, Spain | River Dominated Estuary | 2m | 18.7 ± 33.6 | - | 20.6 ± 24.3 | - | Sierra *et al.*, 2020 |
| Guadalquivir Estuary, Gulf of Cadiz, Spain | River Dominated Estuary | 2m | 0.3 ± 0.5 | - | 6.7 ± 0.4 | - | Sierra *et al.*, 2020 |
| Guadalquivir Estuary, Gulf of Cadiz, Spain | River Dominated Estuary | 2m | 0.9 ± 21.6 | - | 7.3 ± 15.4 | - | Sierra *et al.*, 2020 |
| Noosa River Estuary, Eastern Australia | River Dominated Estuary | 0.5m to 9.6m | -14.24 ± 14.02 | -57.72 to 22.20 | 6.99 ± 0.43 (97 ± 2.2) | 5.92 to 7.95 (90 to 103) | Wells *et al.*, 2018 |
| Mooloolah River Estuary, Eastern Australia | River Dominated Estuary | 0.5m to 6.8m | -7.33 ± 7.25 | -48.76 to 16.31 | 6.74 ± 0.64 (97 ± 3.8) | 5.19 to 7.71 (82 to 112) | Wells *et al.*, 2018 |
| Maroochy River Estuary, Eastern Australia | River Dominated Estuary | 0.5m to 8.2m | 51.33 ± 55.3 | -34.94 to 179.64 | 8.4 ± 1.50 (113 ± 16.7) | 6.07 to 12.93 (92 to 163) | Wells *et al.*, 2018 |
| Pine River Estuary, Eastern Australia | River Dominated Estuary | 0.5m to 10.1m | 17.10 ± 39.44 | -33.22 to 145.50 | 7.1 ± 0.76 (102 ± 6.24) | 6.05 to 8.57 (93 to 117) | Wells *et al.*, 2018 |
| Brisbane River Estuary, Eastern Australia | River Dominated Estuary | 0.5m to 23.9m | 209.54 ± 143.59 | 15.42 to 662.62 | 9.8 ± 1.36 (133 ± 9.9) | 6.75 to 12.75 (105 to 158) | Wells *et al.*, 2018 |
| Middle Reach, Brisbane River Estuary, Eastern Australia | River Dominated Estuary | Surface | 14.5 ± 1.19 | 5.4 ± 0.34 to 25.2 ± 1.87 | - | 13.1 to 17.9 (160 to 250) | Sturm *et al.*, 2017 |
| Lower Reach, Brisbane River Estuary, Eastern Australia | River Dominated Estuary | Surface | 6. ± 0.51 | 3.7 ± 0.85 to 9.1 ± 1.19 | - | 9.2 to 12.7 (125 to 410) | Sturm *et al.*, 2017 |
| Oxley Creek, Eastern Australia | River Dominated Estuary | 2.1m to 13.1m | 210.59 ± 60.23 | 91.54 to 280.16 | 11.7 ± 1.34 (156 ± 19.7) | 9.65 to 14.89 (139 to 199.7) | Wells *et al.*, 2017 |
| Nerang River Estuary, Eastern Australia | River Dominated Estuary | 0.5m to 6.8m | -0.62 ± 20.87 | -67.98 to 45.92 | 6.73 ± 0.43 (100 ± 4.3) | 5.99 to 7.79 (88 to 109) | Wells *et al.*, 2018 |
| Logan River Estuary, Eastern Australia | - | 0.5m to 14.4m | 110.00 ± 153.55 | -54.48 to 796.00 | 9.3 ± 2.36 (127 ± 27.5) | 5.54 to 14.8 (81 to 191) | Wells *et al.*, 2018 |
| Albert River Estuary, Eastern Australia | - | 1.1m to 15.7m | 90.05 ± 73.32 | -9.50 to 264.25 | 10.10 ± 2.24 (131 ± 29.8) | 7.32 to 15.1 (98 to 205) | Wells *et al.*, 2018 |
| Darwin Creek, Australia | Mangrove Creek | ~1m | -0.12 | - | 6.3 (98.9) | 6.0 to 6.8 (95 to 104) | Maher *et al.*, 2016 |
| Hinchinbrook Creek, Australia | Mangrove Creek | ~1m | -3.43 | - | 6.1 (83.3) | 5.6 to 6.8 (75 to 91) | Maher *et al.*, 2016 |
| Melbourne Creek, Australia | Mangrove Creek | ~1m | -1.33 | - | 7.9 (96.6) | 6.9 to 9.1 (86 to 115) | Maher *et al.*, 2016 |
| Morton Bay Creek, Australia | Mangrove Creek | ~1m | -3.19 | - | 5.1 (77.4) | 3.4 to 6.6 (50 to 105) | Maher *et al.*, 2016 |
| Seventeen Seventy Creek, Australia | Mangrove Creek | ~1m | -1.75 | - | 7.7 (94.3) | 7.1 to 8.9 (88 to 106) | Maher *et al.*, 2016 |
| Brisbane River, Australia | - | - | - | - | (285) | (135 to 435) | Musenze *et al.*, 2014 |
| Coffs Creek, Australia | - | - | - | - | (219 ± 37) | (53 to 386) | Reading *et al.*, 2017 |
| Coffs Creek, Australia | - | - | - | - | (266.5 ± 128) | (86 to 678) | Reading *et al.*, 2020 |
| Boambee Creek, Australia | - | - | - | - | (197.1 ± 75) | (87 to 329) | Reading *et al.*, 2020 |
| Bonville Creek, Australia | - | - | - | - | (183.7 ± 65) | (78 to 310) | Reading *et al.*, 2020 |
| Pine Creek, Australia | - | - | - | - | (194.1 ± 65) | (79 to 382) | Reading *et al.*, 2020 |
| Yarra River, Australia | Salt Wedge Estuary | - | - | - | (135.9 ± 31) | - | Tait *et al.*, 2017 |



**7.  Acknowledgments**
We would like to thank GEOMAR for providing the facilities and training (thank you Lea, Florian, and
Chukwudi) required to analyse $N_2O$ samples. We want to thank Torsten and Leonie Schwoch for their sampling
assistance and tireless vessel operation on the Harbour. We want to thank the ADS Environmental Services *Sdn.*
*Bhd.* technical staff for helping to collect portions of this dataset (Grace, Shukry, Atika, Chance, Gene, and
Azza). We would also like to thank our families for supporting our long days away from home.

This research has been supported by internal funding from ADS Environmental Services, Swinburne University
of Technology student travel grant, and GEOMAR.

We have used some of the data available in the MEMENTO database. The MEMENTO database is administered
by the Kiel Data Management Team at GEOMAR Helmholtz Centre for Ocean Research Kiel. The database is
accessible through the MEMENTO webpage: https://memento.geomar.de.





## 8. Data Availability

This data set is available upon request

## 9. Author Contributions

**Johnathan Daniel Maxey** – *Conceptualization, Field Collection, Analytical Methodology, Data Analysis, Writing – Original Draft, Writing – Review & Editing*

**Neil David Hartstein** – *Conceptualization, Field Collection, Analytical Guidance, Writing – Review & Editing, Funding*

**Hermann W. Bange** – *Conceptualization, Analytical Methodology, Data Analysis, Writing – Review & Editing*

**Moritz Müller** – *Conceptualization, Field Collection, Analytical Guidance, Writing – Review & Editing*

## 10. Competing Interests

HWB serves on the editorial board for Biogeosciences. The authors declare that they have no other conflicts of interest.

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
