# Peer review of "NITROUS OXIDE (N2O) in MACQUARIE HARBOUR, TASMANIA"

_EGUsphere, 2024_

## Author Response (AR1)

**RC1 REPLY FROM AUTHORS**

**On behalf of the authors I would first like to thank Reviewer 1 for the thorough review of this manuscript. The feedback provided was constructive and incorporating the reviewer's comments and recommendations will not only ultimately improve the quality and presentation of the work but also allows us to better articulate how we think this system works.**

**Our responses to the reviewer's comments will be posted in-line in bold text below.**

This manuscript reports observations of nitrous oxide concentrations and fluxes from a fjord-like estuary in Tasmania, Australia. This is the first nitrous oxide flux quantification study from this estuary and, to the best of my knowledge, Tasmania. The authors found that the deeper waters within the harbour were a constant source of nitrous oxide, but the air-water fluxes were largely regulated by an allochthonous freshwater lens that was undersaturated in nitrous oxide.

The manuscript constrains endmember inputs into the harbour well, but internal processes within the harbour could be investigated further, particularly in regard to inorganic nitrogen dynamics and any potential influences arising from an active aquaculture industry (salmon farming) within the harbour that is currently ignored by the study. Uncertainty also needs further clarification (see comments below). I recommend the manuscript for publication, however there are some issues that will need to be addressed before publication. I have provided general and specific comments below.

> **We thank the reviewer for these comments and agree that the effects of commercial aquaculture on water column $N_2O$ dynamics is an interesting topic worthy of exploring further, especially in this system. Quantifying the effects of an aquaculture farms on water column processes (including $N_2O$ dynamics) can be challenging especially relative to the more easily detectable effects on the seabed (which are well documented).**
>
> **We are currently preparing a separate contribution, designed to address the question of farm impacts to the water column. Previous studies investigation the impact of farms on oxygen consumption found that they were extremely near-field and not detectable away from the farm (Maxey *et al.,* 2020). We used a more targeted experiment design to address the question of farm impacts to water column $N_2O$ dynamics (*in prep*). For this current manuscript we will offer some speculation and insights as to how the industry may be affecting this system and under what conditions this might occur.**
>
> **We will improve the reporting of uncertainty as suggested, including clarifying how uncertainty was propagated and accounted for in the flux calculations.**

**General comments:**

Writing quality throughout the manuscript is not quite up to a publishable standard. The manuscript is generally well written, but requires more editing and polishing to improve readability (mostly in results and end of discussion) and to fix grammatical errors. Some of the paragraphs towards the end of the discussion and conclusion could also be combined (I suspect this could be an upload issue). I have highlighted some issues but not all in the manuscript.

There appears to be minimal quantification or discussion of uncertainty throughout the manuscript. Was the uncertainty propagated throughout the flux calculations or is the uncertainty indicated just the standard error between the samples?

> **The uncertainty shown in the Air/Sea flux table represents standard error of the mean of the individual Air/Sea flux replicate estimates. Replicate Air/Sea flux estimates were computed using replicate concentration measurements.**

> **We will amend these figures to report uncertainty in terms of standard deviation to better align with how others have chosen to report error (*see* Sierra *et al.,* 2020; Zhang *et al.,* 2020; Wang *et al.,* 2023). In addition, we will propagate uncertainty using methods described in Ku (1966).**

> **We will also add standard deviation values to Table 3 as well.**

It looks very low as there is usually a high degree of uncertainty in flux calculations/estimations. Error needs to be propagated throughout most calculations, and how it is propagated should also be indicated somewhere in the methods.

> **We will amend the manuscript as suggested. This appears to be an artefact of both originally reporting error as standard error and a need to improve error propagation (see above). Our mean relative error for the $N_2O$ concentration values are provided in section 2.5.1 and are similar to values reported in Bange *et al.,* (1996, 1998) and Sierra *et al.,* (2020).**

> **An additional clarification for the presentation and calculation of uncertainty will be added in our results tables to section 2.5.2.**

Uncertainty in the study design should also be discussed, while I believe the study design to be overall very good, the sampling stations in the harbour are all through the deep central complex, which may or may not be representative of the entire harbour.

> **We agree that there is some methodological uncertainty, especially in the absence of measured gas transfer velocities on site. However, we have tried to constrain this by presenting Air/Sea Flux estimates using a range of published k600 parameterizations that are appropriate for Macquarie Harbour waters (see Raymond and Cole 2001). We will also add some open ocean k600 parameterizations as well to help bracket those values (*e.g.* Nightingale *et al.* 2000 and Wanninkhof 2014).**

> **Our sampling design was focused on providing a description of end-member conditions, surface waters, and basins of the harbour through its main axis, where the majority of the water is, and where residence time is longest. This design was not focused on capturing $N_2O$ distribution around the farms, which will be addressed in a separate study. What this study does show is a heterogenous distribution and air/sea flux in the surface waters even within the same sampling period.**

> **Some additional commentary clarifying the limitations of this study (in terms of Harbour representation) will be add to the final manuscript.**

There is an active aquaculture industry (presumably salmon farming) within the harbour, which would contribute nutrients and organic matter directly into the harbour and could directly or indirectly influence nitrous oxide dynamics. However, there is no mention or discussion anywhere in the manuscript apart from the outlines of fish farming leases in Figure 1. Why is there no mention or discussion of these aquaculture activities in the manuscript?

**The aquaculture farms shown in figure 1 are open sea cage salmonid farms that are suspended from the surface down to a depth of about 15m. We agree that the effects of salmonid aquaculture on water column N$_2$O dynamics should be investigated further, which is presently a significant knowledge gap not only for this study but for any investigation of fjord like estuaries or aquaculture sites in the wider literature.**

**We are preparing a follow-up contribution that will help address this knowledge gap in the harbour and wider literature in general. Stay tuned.**

Why is the inorganic nitrogen data from the measuring stations within the harbour not presented? The endmember inputs of inorganic nitrogen (and N$_2$O) are constrained and displayed very well, and I feel the manuscript would benefit greatly by displaying this data as there appears to be an active sub-halocline N cycling/N$_2$O source within the harbour.

**Thank you for this comment, we will add inorganic N data to figure 3. Note that correlations presented in figure 5 already include these data, so we do not expect showing these data to influence the final results of this study.**

The manuscript claims in places that the harbour is sequestering or acting as a sink of N$_2$O when it receives high inflows from the Gordon River. However, the manuscript presents no evidence and the data does not support these claims. The estuary is always supersaturated in N$_2$O in the deeper sub halocline waters and there is always a positive diapycnal flux (indicative of a source). While the undersaturated freshwater lens (originating from the endmember rivers) will uptake N$_2$O from the atmosphere and deeper waters (diapycnal flux), there is no evidence presented that N$_2$O is being 'sequestered' or denitrified within the surface waters and it is more likely the surface waters are exporting any collected N$_2$O to the ocean.

**Thank you for this comment, we agree that the term "sequester" is misleading and should be removed from the manuscript. We adjust the text to discuss the fate of N$_2$O in terms of transport as this more accurately represents what we think is occurring in this system.**

**Specific comments:**

**Thank you for these comments. They have helped us add clarity to the manuscript. We have responded to each of them below.**

Line 13: I find 'Cruises' is a term normally used for high-resolution spatial surveys with logging equipment. This study used a long-term time-series approach at fixed sampling sites/stations.

**For additional clarity we will replace the term "cruises" with "sampling surveys" as suggested.**

Line 53: Although far less common, there are also abiotic (chemical processes) that may produce N$_2$O, please clarify.

**We will amend this section to use the more precise term "Biological N$_2$O production" instead of "N$_2$O production" as suggested.**

Line 54: I suggest 'influenced' may be a better word than 'governed'.

**We will replace the term "governed" with "influenced" as suggested.**

Line 55: 'Inorganic nitrogen availability' would be more appropriate than just ammonium because nitrate and nitrite are also important.

**We will replace the term with "Inorganic nitrogen (N) availability" as suggested.**

Line 58: I recommend inserting 'often' or a similar word before 'have disproportionately high biological productivity', because I am sure the reverse can be true for some systems.

**We will add the term "often" as suggested.**

Line 63: Clarify this statement as water/atmospheric concentration gradient is also a factor controlling air-sea fluxes. One way to clarify this statement would be to insert 'physical' before 'factors controlling' and then change 'waterbody/atmospheric concentrations' to 'water/atmospheric concentration gradients, current velocities, depth, and wind speeds'

**We will modify the passage by incorporating the suggested terms and to break it into two sentences to read:**

**[…*depending on physical drivers of air/sea fluxes including waterbody/atmospheric concentration gradients, current velocities, depth, and wind speed (Wells et al., 2018; Bange et al. 2019). Other factors include land use modification (Reading et al., 2020; Chen et al., 2022) and even the presence of microplastics (Chen et al, 2022).*]**

Lines 89 to 103: I find this paragraph to be more discussion than methodology, plus a lot of the information is repeating information in the preceding paragraph. Some of this could be used in discussion to explain nitrous oxide dynamics or remove any duplicated information.

**We will restructure this passage, remove repeated information, but will retain information describing the study site's poor water clarity and water residence time estimations.**

Line 96: Doesn't the 'low catchment rainfall' statement contradict the statement on line 109, where it states the catchment receives high rainfall?

**This region receives the greatest rainfall volume in Australia, but there are still seasonally high rainfall periods, which have been shown to affect river flow and frequency of marine intrusions.**

Lines 105 to 113: What influence does the dam on the Gordon River play on hydrology/discharge into the harbour?

**Unfortunately, there are no robust hydrological models available for this system that include coupled surface and ground water processes. Our rainfall/runoff calculations suggest that the proportion of Gordon Flow that is passing through the GAD flow gauge (better representation of dam release) can be as high as about 28% (July 2022).**

**Dam release water may also have properties differing from the runoff water entering the harbour that can affect harbour DO and $N_2O$ dynamics as well such as high particle loads, high organic matter concentrations, low dissolved oxygen, and low $N_2O$ concentrations. Unfortunately, this has not been investigated in this system.**

Line 161: Should 'River loading' be 'River discharge' or 'river discharge for loading calculations'? please clarify.

**We will amend the passage to use the term "river discharge" instead of "loading" as suggested.**

Line 212: The equation for %$_{sat}$ should be the $C_{obs}$ **divided** by $C_{eq}$ before multiplying by 100.

**We will amend the equation to use more precise notation as suggested.**

Line 229: Consider replacing 'analysed' with 'determined', which is more appropriate for describing statistics. Statistical tests are only a part of scientific analysis.

**We will amend the terms as suggested.**

Line 229: 'The effects of' is not what an ANOVA test is used to test, it should be 'Differences between'.

**We will amend the passage to use the term "differences between" as suggested.**

Line 238: Were statistics used to check for these differences?

**We will amend this passage and Data Analysis Section to better describe the statistical methods as well as incorporate those statistics into the the results.**

Lines 259 to 260: Isn't this site also close to the fish farm leases? I would assume they are potentially a significant nutrient source that could impact oxygen demand.

**Site WH2 is approximately 3 kilometers upstream of the nearest farm lease. Previous work by Maxey *et al.* (2020) investigated the influence of this lease on water column oxygen demand at this region of the harbour and found that the farm's influence was constrained to the water near the seabed and that even this was confounded by the presence of the Gordon River plume.**

**Other important factors to consider include this site being the furthest from marine intrusions, the water column in the basins being isolated by the presence of a halocline, poor light availability below the halocline, and the basin water residence time being the longest in the harbour.**

**There was little to no evidence in previous numerical modelling studies that indicate that nutrient release and debris plumes migrate that far upstream.**

Line 295: The trend line should be removed from the TAN subset (c) as the relationship is not significant. I also recommend including dissolved oxygen concentration as the negative relationship is quite strong.

**We will amend the figure as suggested.**

Lines 313 to 314: I am not sure that 'sink' is the right word; I think up-taking N$_2$O from the atmosphere would be more accurate (see comment for lines 350 to 352).

**Thank you for this suggestion. We will amend the text to use more precise language as you suggest. Sink would imply that the N is being removed from the system as N$_2$ gas, but instead this N is being transported to the surface layer.**

Line 339: What is the uncertainty in these estimates presented in Table 3?

**We will amend the table as suggested to include metrics of uncertainty.**

Line 350 to 352: I am not sure I agree with this statement that the harbour stations were a sink and I think it needs to be clarified. A 'sink' would indicate the nitrous oxide is up-taken and reduced to $N_2$ (most likely by denitrification where microbes would reduce it to $N_2$). While it appears the surface lens emulating from the Gordon River is uptaking atmospheric nitrous oxide, the high sub-halocline nitrous oxide concentrations and positive diapycnal flux across all seasons would indicate the sub-halocline waters and or sediments are still producing nitrous oxide. I suspect the allochthonous surface lens is just laterally exporting internally produced nitrous oxide from the harbour.

**We thank the reviewer for this insight. Moving forward we will amend the passages as suggested to more precisely describe the $N_2O$ dynamics as transport mechanisms rather than transformative mechanisms.**

Lines 355 to 358: This is almost a separate topic and could benefit from more discussion linking the discharge to $N_2O$ flux dynamics in the harbour. Also, while the dam release definitely contributes to Gordon River flow, at its peak contribution, it made up just over one-quarter of the flow/discharge from the river (72% or the flow was from catchment rainfall). I think the links between dam release/rainfall and harbour $N_2O$ fluxes need to be clarified and discussed further.

Line 355: I don't think the freshwater is a 'key driver' of nitrous oxide emissions, it does not appear to be causing or preventing nitrous oxide production, rather it appears to be laterally transporting nitrous oxide from the harbour. However, it does appear to regulate nitrous oxide air-sea exchange in the harbour.

**We would like to clarify that emission is a physical phenomenon, and production / consumption is primarily a biological one in this system. The harbour seems to be continuously producing $N_2O$ in the basins, yet it is not always emitting them to the atmosphere, due to physical drivers.**

**The mechanism controlling the air/sea flux (emission) of the $N_2O$ produced in the basins appears to be river flow in this system. We suspect that an undersaturated surface lens flushing through the harbour captures $N_2O$ in its dissolved form and transports it to the ocean in its dissolved form, preventing its emission to the atmosphere. Freshwater flow explains approx. 64% to 81% of our observed surface flux.**

Lines 393 to 395: Wouldn't the positive relationship with nitrate also support this assumption? i.e., as the end product of nitrification.

**We thank the reviewer for this insight and will amend the passage to also reference the relationship between $NO_3^-$ and $N_2O$**

Lines 452 to 453: This statement is not supported by the data presented in this study. There is no evidence that $N_2O$ is being sequestered within the harbour. If $N_2O$ was being sequestered within the harbour, then the sub-halocline waters would also be undersaturated. The Gordon River (endmember) is likely sequestering $N_2O$ as indicated by the undersaturated waters discharging into the harbour, but there appears to be a net $N_2O$ source within the harbour all year round.

**We thank the reviewer for this insight and will amend the passage to use language focused on transport and uptake mechanisms rather than using the term sequester (as this implies a net removal of $N_2O$ via transformation in to $N_2$).**

**Technical corrections**

**We will amend the manuscript to address these corrections.**

Line 21: Define the acronym for AOU before using the acronym – Apparent Oxygen Utilization

Line 36: Insert a comma after worldwide.

Line 42: Define the acronyms for 'NAO' and 'SAM' before using the acronyms

Line 46: Define the 'GHG' acronym, as this is the first use.

Lines 46 to 47: Please clarify the sentence as it is awkward as written, especially the last part. It may be best to break it into two or more sentences, with the first part stating that 'anthropogenic emissions' are responsible for increased atmospheric molar fractions. Followed by the specific sources. Additionally, including the 'marine emissions' in a stand-alone sentence will also lead into the discussion later in the paragraph.

Line 50: Insert a comma after the references.

Line 50 to 53: I recommend constructing a new sentence regarding the ozone layer depletion as it is awkward as written because it is a separate subject in an already busy sentence.

Line 76: Include 'Australia' in the location description.

Line 80: Define the acronym for DO as this is the first use.

Lines 83 to 84: Please clarify this sentence to indicate the concentrations refer to dissolved oxygen.

Line 100: Insert a comma after model.

Line 108 to 109: The reference should be at the end of the sentence.

Line 125: Check with the journal conventions; use the full word instead of an acronym to start the paragraph.

Line 131: Insert a comma after depth.

Line 141: The acronym for N should be defined here as this is the first appearance, I see it is defined later at line 151.

Line 151: Should the acronym be DIN in this line? If not, it should be removed (see comment for line 141).

Lines 174 to 175: This sentence should be at the end of the preceding paragraph.

Line 192 to 220: There is inconsistent use of italicisations of the equation parameters throughout.

Line 238: Insert a comma after metrics.

Line 245: Insert a comma after TAN loading if this sentence is retained as is.

Line 248: Patterns of $N_2O$ loading are very vague; what were the ranges of these loadings?

Lines 304 to 305: Fix this reference, most likely an upload error.

Line 324: remove the space between the left bracket and smaller.

Line 327: The M in μmol should be lowercase.

Lines 346 to 348: Should these two paragraphs be one?

Line 365: Insert a comma after the reference.

Line 367: Insert 'and' before 'low $NO_3^-$ concentrations.'.

**RC2 REPLY FROM AUTHORS**

On behalf of the authors I would first like to thank the Reviewer 2 for the thorough review of this manuscript. The feedback provided was constructive and incorporating the reviewer's comments and recommendations will not only ultimately improve the quality and presentation of the work but also allows us to better articulate how we think this system works.

**Our responses to the reviewer's comments are posted in bold text below.**

Major comments:

1. It is unclear what explains the trend in shallow freshwater lens across the bay. One would expect a trend of moving towards equilibrium in the July campaign as water transits along the bay due to simple gas exchange (and also any transport from below the halocline). Yet the data (as far as I can tell) does not show this? Do the authors have an explanation for the spatial distribution of N2O in the freshwater lens?

> **Thank you for the insightful question. Freshwater inflows were greatest in July 2022 as seen in Figure 2 and 3 (approx. 381 cumecs). During this period the surface lens residence time is lowest *i.e.* moving through the harbour rapidly. Note that some water will enter the harbour through the King River located at the northeast end of the harbour and this water will mix with Gordon River water and aid in the advection of water out of the harbour. We suspect that the residence time of the surface lens during this period does not allow for enough Air/Sea flux to result in saturating the surface waters with N$_2$O.**

2. A follow up from the previous comment – what is the residence time of the freshwater lens within the harbour? If known then perhaps a mass balance approach might be useful to look at which may help answer the comment above.

> **Unfortunately, water residence times in this system are still poorly constrained, but oxygen tracer modelling presented in Andrewartha and Wild-Allen (2017) (see original manuscript for full reference) show tracer concentrations were reduced by half in approximately 40 days at the surface under "normal" flow conditions. July 2022 represents relatively high flow conditions for this system and may have reduced water residence times to just a few days.**

3. The choice of k model – I suspect that the use of Raymond and Cole 2001 would result in an overestimate, simply because of where the model was developed (shallower estuaries than Macquarie Harbour). Might be worth considering incorporation of some ocean parameterization (e.g. those of Ho or Wanninkhof) to at least bracket some of the flux estimates.

> **Thank you for this suggestion. We will incorporate additional k parameterizations to the flux estimates. We have chosen to incorporate those presented in Nightingale *et al.* (2000) and Wanninkhof (2014) as they were developed under similar wind conditions as those observed in the harbour.**

4. I am not sure the AOU vs deltaN2O relationship provides clear evidence for a nitrification only source. Denitrification could also play a role there, with AOU and deltaN2O simply covarying with residence time of bottom waters.

> **Thank you for this comment. We agree that denitrification could also be influencing N$_2$O concentrations in this system, both as a sink for N$_2$O or as a source. The oxygen concentrations observed during this sampling period did reach single digit micromolar**

**concentrations at one of our sites (This then permits N$_2$O production via denitrification); These conditions are not uncommon in this system (see Maxey *et al.* 2022).**

**Given that the majority of the water column is normoxic and that there were linear relationships between AOU and deltaN$_2$O and [NO$_3^-$] and N$_2$O saturation, the predominant water column driver of N$_2$O production is still likely nitrification. Nevertheless, you make an excellent point and we will clarify that denitrification may also be an important driver in certain portions of the harbour's water body and under certain conditions.**

5.    The undersaturated N2O in the upstream endmember, and freshwater lens is a very interesting observation. I think some further explanation as to the potential mechanisms of this would be useful. Have there been N2O measurements in the Gordon River upstream of the Harbour? I do not think the release of dam water would impact the observed undersaturation at the inlet of the Harbour, simply because the transit time from the dam to the Harbour would be much longer than the reaeration time for the river.

**We have suggested that the main driver of N$_2$O undersaturation is denitrification along the riverbed and supported this with literature starting on Line 363. Complete denitrification would result in N$_2$O uptake. Unfortunately, like Macquarie Harbour, water column N$_2$O concentrations in the Gordon River have not been studied previously.**

Minor comments:

Lines 48 – 51 – The order of statements here is a little weird with the GWP of N2O being linked to ozone depletion capacity of N2O. Would separate into 2 distinct statements.

**We agree that this section could use some streamlining. The paragraph will be restructured with focus being placed on GWP, mechanisms of production, and environmental drivers of N$_2$O dynamics.**

Line 109 Remove "volume"

**We will make the suggested amendment.**

Fig 4 is a little hard to digest – can you add a curtain/contour plot for each sampling period which would make interpretation of spatial trends a little easier.

**We will incorporate contour plots focusing on conditions through the main axis of the system for both N$_2$O and dissolved oxygen.**

Line 304 Something wrong with citation there

**We will amend the passage.**

---

## Author Response (AR2)

**We thank the Reviewer for their thorough work examining this revised manuscript. We are pleased to address the technical corrections presented below. We will respond to each in bold text.**

*Technical corrections*
*(line numbers refer to the manuscript, not the 'authors tracked changes' version)*

*Line 16: Please indicate at what depth the end member samples were collected and clarify that the samples at the harbour stations were throughout the water column.*

**We have added text to specify the sample collection depths in the abstract as requested. The amended passage starting on line 16 now reads:**

> **"$N_2O$ samples were collected from mid water depths at the ocean (5m) and minor river (1m) endmembers, 2m from the bottom (10m) at the major river endmember, and at 5 depths through the water column at 4 stations within the main harbour body."**

*Line 18: Replace is with 'was' (past tense).*

**Passage amended as requested.**

*Line 81: Insert a space after the reference before the sentence.*

**Passage amended as requested.**

*Line 214: Insert 'were' before determined.*

**Passage amended as requested.**

*Line 327: Fix the reference.*

**Reference amended as requested.**

*Line 367: Please clarify that the endmembers you are referring too in this sentence are rivers.*

**Passage amended as requested. The amended passage starting on line 367 now reads:**

> **"Our observations of river endmember $N_2O$ concentrations were similar to the lower end of the concentrations reported in…"**

*Line 371: Replace 'denitrifies' with 'denitrifiers'.*

**Text amended as requested.**

*Line 384: Should this be Figure '5' and Figure '6'?*

**References amended to specify the correct figure number.**

*Lines 396 to 397: Please correct the figure referrals, which should be Figure 7C and Figure 7D.*

**Figure references corrected as suggested.**

*Line 420: Fix the reference.*

**Figure references corrected as suggested.**

*Line 431: Add in the space between 'laterally and out'*

**Passage amended as requested.**

*Line 434: Indicate which figure you are referring to (Figure ?)*

**Figure references corrected as suggested.**

*Line 438: Indicate which figure you are referring to (Figure ?)*

**Figure references corrected as suggested.**

*Lines 485 to 489: These sentences would read better either incorporated into the previous paragraph or at the end of the paragraph at lines 468 to 476 where climate change is addressed.*

**Passage amended as requested. We have chosen to incorporate these sentences into the end of the climate change paragraph starting on line 468. It now reads:**

> *"Climate change predictions for Tasmania's West Coast (which includes the Macquarie Harbour catchment) indicate that the region will experience a more extreme precipitation regime with increased winter precipitation and decreased summer precipitation (Grose et al., 2010; Bennett et al., 2010). If these future predictions result in more extreme seasonality in Gordon River flow, then the harbour may respond in kind with a larger variation in $N_2O$ air / sea flux i.e. greater $N_2O$ atmospheric uptake in winter and greater $N_2O$ emission in summer. However, given that the river flow is somewhat regulated by the hydroelectric dam, our study suggests that flow regulation has the potential to augment harbour $N_2O$ emissions. Releasing water during extreme low rainfall periods might allow $N_2O$ slowly accumulating in subhalocline waters to be released in the exported surface lens. Fjord and fjord-like estuaries are defined by their strong stratification and sensitivity to freshwater inputs. With climate change, rainfall patterns are expected to become more extreme and thus alter the river flow, and subsequently $N_2O$ source sink dynamics in these systems on a global scale. In systems that are expected to experience increasingly drier conditions they may shift from net sinks of $N_2O$ to sources, and further perpetuate the accumulation of $N_2O$ in the atmosphere."*